# Sex Differences in Tryptophan Metabolism: A Systematic Review Focused on Neuropsychiatric Disorders

**DOI:** 10.3390/ijms24066010

**Published:** 2023-03-22

**Authors:** Mariana Lapo Pais, João Martins, Miguel Castelo-Branco, Joana Gonçalves

**Affiliations:** 1Doctoral Program in Biomedical Engineering, Faculty of Sciences and Technology, University of Coimbra, 3000-548 Coimbra, Portugal; 2Coimbra Institute for Biomedical Imaging and Translational Research (CIBIT), University of Coimbra, 3000-548 Coimbra, Portugal; 3Institute for Nuclear Sciences Applied to Health (ICNAS), University of Coimbra, 3000-548 Coimbra, Portugal; 4Faculty of Medicine, University of Coimbra, 3000-548 Coimbra, Portugal

**Keywords:** tryptophan metabolism, serotonin, kynurenine, sex, gender, neuropsychiatric disorders

## Abstract

Tryptophan (Tryp) is an essential amino acid and the precursor of several neuroactive compounds within the central nervous system (CNS). Tryp metabolism, the common denominator linking serotonin (5-HT) dysfunctions and neuroinflammation, is involved in several neuropsychiatric conditions, including neurological, neurodevelopmental, neurodegenerative, and psychiatric diseases. Interestingly, most of those conditions occur and progress in a sex-specific manner. Here, we explore the most relevant observations about the influence of biological sex on Tryp metabolism and its possible relation to neuropsychiatric diseases. Consistent evidence suggests that women have a higher susceptibility than men to suffer serotoninergic alterations due to changes in the levels of its precursor Tryp. Indeed, female sex bias in neuropsychiatric diseases is involved in a reduced availability of this amino acid pool and 5-HT synthesis. These changes in Tryp metabolism could lead to sexual dimorphism on the prevalence and severity of some neuropsychiatric disorders. This review identifies gaps in the current state of the art, thus suggesting future research directions. Specifically, there is a need for further research on the impact of diet and sex steroids, both involved in this molecular mechanism as they have been poorly addressed for this topic.

## 1. Introduction

Tryptophan (Tryp) is an essential amino acid ubiquitously found in living cells [1,2]. Absorption and metabolism of Tryp occur along the gut–brain axis. Once absorbed from the gut, Tryp becomes available in the circulation for distribution to target sites both peripherally and centrally [2,3,4,5]. In addition to endogenous Tryp metabolism, resident gut microbiota also contributes to the production of specific Tryp metabolites and indirectly influences host physiology [6].

This organic molecule is also the precursor of several biologically essential compounds produced along two critical pathways: hydroxylation to produce serotonin (5-HT) and oxidation to produce kynurenine and its metabolites (Figure 1). The regulation of Tryp concentration is vital to maintain systemic homeostasis, as both paths have different and fundamental neurological roles in the organism [7]. On one hand, 5-HT signalling is involved in the physiological regulation of behavioural, neuroendocrine, and higher brain functions, such as cognition and emotional states [7,8]. So, deficient 5-HT levels may interfere with several physiological conditions. On the other hand, the kynurenine pathway is essential for intestinal homeostasis and the regulation of immune responses [9,10,11,12,13]. Furthermore, several kynurenine downstream metabolites have neuroactive functions, such is the case of quinolinic acid (QUIN) and kynurenic acid (KYNA), which have both neurotoxic and neuroprotective effects depending on their levels [4,14,15,16]. Under normal conditions, local concentrations of KYNA and QUIN are low. However, upon an insult, QUIN production is partially overactivated due to insufficient KYNA concentration to block its production [14,17]. Some studies have demonstrated that modulation of the kynurenine pathway by enhancing KYNA or decreasing QUIN can be a potential therapeutic strategy in neurological disorders [10].

It is well described that Tryp metabolism is impaired in a wide range of disorders, such as neurologic/psychiatric conditions [1,7,9,10,14,18,19,20]. Interestingly, most of those conditions appear in a sex-specific manner [21]. Generally, women are more vulnerable than men to most psychiatric disorders that affect emotions, including major depressive disorder (MDD) and a host of anxiety-related conditions, such as generalised anxiety disorder, panic disorder, post-traumatic stress disorder, and phobias [21,22,23]. Conversely, neuropsychiatric conditions such as schizophrenia, attention-deficit/hyperactivity disorder (ADHD), autism spectrum disorder (ASD), and substance/drug abuse disorder occur predominantly in men [21,22,24]. Interestingly, that sexual bias seems to be related to opposite disturbances in the 5-HT system. On one hand, most neuropsychiatric disorders with aggravated manifestations in women (e.g., mood disorders) are associated with deficient production of 5-HT [25,26]. On the other hand, neuropsychiatric conditions that affect more males than females (e.g., ASD and ADHD) have excessive production of that neurotransmitter [10,24,27]. Sex differences are also seen in neurodegenerative and autoimmune disorders, as the male sex is a significant risk factor for Parkinson’s disease (PD) and motor neuron disease, whilst females are more susceptible to Alzheimer’s disease (AD) and multiple sclerosis [22,28]. Exceptions of sexual dimorphism in prevalence are bipolar disorder and obsessive-compulsive disorder (OCD), which report an almost equal gender ratio [21]. Sex can potentially affect disease process and progress through differences in chromosomal complement, gene expression, hormones, organs, and various physiologic processes [28], and so, it is not surprising that women and men have different incidences and disease manifestations [28], which remains to be disclosed.

Here, we will discuss the current understanding of sex differences on Tryp metabolism and how they may influence the onset and progression of neuropsychiatric diseases. This review also highlights important gaps in the current state of the art regarding sexual dimorphism on Tryp metabolism, thus suggesting future research directions.

### 1.1. Sex Differences in Tryptophan Metabolism

Under normal conditions, sexual dimorphism in Tryp metabolism has been suggested. Specifically, animal studies demonstrated that females display higher circulating levels of Tryp compared with males [29,30,31], which is also reflected by an increase in 5-HT production [32,33]. Moreover, female rats exhibited increased levels of Tryp [34,35] and 5-HT levels in several brain regions [35,36,37]. Interestingly, one preclinical study showed a sex-specific role for the microbiota in regulating central nervous system (CNS) 5-HTergic neurotransmission profiles in which the majority of central effects of germ-free status were present only in male animals [38]. Findings obtained from clinical data also reported an increase in circulating levels of Tryp and brain availability of this amino acid (estimated by increased serum ratios of free Tryp to its competing amino acids (CAA)) [5,39] and elevated 5-HIAA concentrations in cerebrospinal fluid samples in women [40,41]. However, not all the studies were consistent. Indeed, some works found no significant sex differences in Tryp and Tryp/CAA ratios [42,43] and even higher plasma Tryp levels in men [44,45]. Although few studies address sex differences in kynurenine metabolites, these are often in disagreement. Some of them described similar kynurenine levels in both sexes [30,33,39], while others found lower KYNA plasma levels in women [17,30].

#### 1.1.1. Hydroxylation Pathway: Tryptophan and 5-HT Synthesis

Consistent evidence demonstrates that healthy women have a more significant serotoninergic system activation. Specifically, compared with males, females have increased Tryp availability and greater 5-HTergic synthesis that extend from circulating levels to brain concentrations (Table 1). In particular, animal studies showed that females had higher peripherical levels of plasma 5-HT [32,33]. Moreover, female rats exhibited elevated 5-HT levels in the brainstem, limbic forebrain, and hypothalamus/preoptic area [35,36,37]. Accordingly, human cerebrospinal fluid studies also suggested that the rate of brain 5-HT metabolism is higher in women [40,41].

#### 1.1.2. Oxidation Pathway: Tryptophan and Kynurenine Metabolites

Concerning the Tryp oxidation/kynurenine pathway, there is a low coherence between the existing studies on sexual dimorphism. Indeed, some evidence points to similar levels between sexes, while others point to lower levels in females (Table 1). Specifically, a preclinical study showed that female rodents had lower plasma concentrations of kynurenine but similar levels of KYNA in plasma and cortical and hippocampal regions compared with males [30]. Others reported no sexual dimorphism in plasma and whole brain kynurenine/Tryp ratios in animal models [33]. In agreement, one human study indicated that both sexes exhibited similar levels of plasma kynurenine and KYNA [39], while another exposed reduced plasma levels of KYNA in women [17]. Although this latter result suggests women may have decreased capacity to produce the neuroactive compound KYNA, more data are needed to validate this assumption.

### 1.2. Impact of Tryptophan Nutritional Status between Sexes

Tryp is obtained exclusively from the diet with an estimated dietary requirement of 5 mg/kg/day [2,46]. Besides serving as a building block for proteins, Tryp is a critical nutrient for the nervous and immune systems’ functions. So, alterations in diet and, consequently, in Tryp availability can cause variations in serotonin, kynurenine, and their downstream metabolites [46]. Eating behaviour, food choice, and nutritional strategy, which can be conditioned by intra-individual (biological or psychological) and extra-individual (socioeconomic and cultural) factors, may lead to changes in this amino acid availability [47]. Evidence shows that diet significantly modulates brain–gut–microbiome interactions with important implications for brain health and pathological brain condition [46]. Unlike Tryp and kynurenine, 5-HT cannot pass the blood–brain barrier (BBB) [7], and central biosynthesis of this neurotransmitter depends entirely on Tryp availability, which, in turn, relies on the ratio to other CAA [5]. Accordingly, Tryp supplementation or depletion through the diet can directly affect 5-HT brain levels and its primary functions [48].

#### 1.2.1. Tryptophan Supplementation

Tryp supplementation has been widely used in basic research to facilitate the entry of Tryp into the brain, and thus elevate 5-HT synthesis and release [48]. The adequate intake of Tryp has been associated with several beneficial outcomes since it is crucial for physical and neuronal growth and development [49]. Several reports have suggested that Tryp supplementation could be an alternative or a co-adjuvant approach for the treatment/amelioration of neuropsychiatric conditions, such as mood, stress, and anxiety disorders, or even ASD, ADHD, AD, and PD [5,48,49,50,51]. However, dose–response experiments with Tryp supplementation in humans also proved that excessive doses could have negative consequences by activating tryptophan-2,3-dioxygenase (TDO) and Tryp oxidation [39]. Moreover, excess Tryp can also lead to Tryp hydroxylase (TPH) inhibition and mood lowering by decreasing 5-HT brain levels [52].

Pieces of evidence reported sex differences in response to the enhancement or lack of Tryp through diet (Table 2). In 1995, Walsh et al. stated that men were less likely to exhibit diet-enhanced alterations in brain 5-HT function than women [53]. Accordingly, it was described that Tryp supplementation caused higher plasma-free Tryp in women, but similar concentrations of kynurenines in both sexes [39]. According to Zahar et al., the effect of Tryp on mood ratings is dependent on sex, with men showing a positive mood increase and women exhibiting a weak negative trend after increasing Tryp to large neutral amino acids (LNAA) [5]. Accordingly, more data revealed that higher levels of Tryp were associated with trait hostility, a propensity for anger, and a tendency to express anger outwardly only in women [54]. This effect contradicts the idea of Tryp administration resulting in positive behavioural changes related to 5-HT increase instead of negative ones. However, as previously pointed out, excessive doses of Tryp could also have negative consequences. So, we can hypothesise that women may be more susceptible to excessive dietary Tryp. Yet, dose–response studies between sexes should be performed to further address this issue.

#### 1.2.2. Tryptophan Depletion

On the other hand, Tryp depletion has been used as a research tool to investigate human serotoninergic processes [48]. Since the synthesis of neural 5-HT depends entirely on Tryp availability [5], not surprisingly, malnutrition or excessive dietary restriction of Tryp decreases brain 5-HT supplies and, consequently, causes behavioural changes related to anxiety, depression, hyperactivity, and impulsivity [49]. Moreover, nutritional physiologists recognised that dietary deficiency of Tryp could increase the neurotoxic metabolites in the kynurenine pathway [49] and result in CNS sequelae, such as ataxia, cognitive dysfunction, and dysphoria [55,56]. Clinical evidence suggests that depletion of this amino acid worsens depressive symptoms and mood states [57]. Additionally, a lack of Tryp increases cognitive dysfunction in people with AD [58]. Once again, sexual dimorphism arises following the alteration of Tryp levels through the diet, where women are more significant affected than men (Table 2).

In particular, a clinical neuroimaging study using in vivo α-[^11^C]methyl-L-tryptophan positron emission tomography (PET) indicated that women exhibited a more significant effect of acute Tryp depletion (ATD) on diminished brain 5-HT synthesis [59]. Moreover, others revealed that the impact of Tryp depletion on verbal memory, mood, and emotion processing was also more prominent in women than in men [60,61,62,63]. All these results support a greater susceptibility of women to suffer serotoninergic alterations through changes in the levels of its precursor.

### 1.3. Effect of Sex Hormones on Tryptophan Metabolism

Sex hormones contribute to sexual dimorphisms mainly in energy balance, metabolic homeostasis, and behavioural traits, including mood and cognitive function [64,65,66,67]. In women, the menstrual cycle is divided into two main stages, the follicular and luteal phases, essentially differentiated by oestrogen and progesterone release, respectively [68]. Those hormonal fluctuations during the women’s menstrual cycle are associated with numerous physiological and psychosocial changes [68], such as the availability of circulating Tryp. Since Tryp metabolism affects the synthesis rate of 5-HT and catabolism through the kynurenine pathway, sex steroids might indirectly modify cerebral neurotransmitter concentrations and, consequently, mood, behaviour, fatigue, stress-coping, and immune responses [17,29,69]. In agreement, several authors have been exploring sex hormones and their influence on Tryp metabolism using in vitro approaches, animal models, and human subjects (Table 3).

Furthermore, increased incidence of depression in women following childbirth (e.g., post-partum depression), as well as during perimenopause and menopause, suggests that disturbances in sex hormones significantly affect women’s susceptibility to depression [22]. Accordingly, fluctuations in hormonal levels during the menstrual cycle have been proven to significantly impact on several neuropsychiatric disorders, such as depression, epilepsy, migraine, seizures, multiple sclerosis, stroke, and PD [28,70,71,72]. Specifically, the periovulatory stage during the follicular phase (oestrogen rising) and the premenstrual stage during the luteal phase (oestrogen and progesterone decline) have been proven to aggravate the frequency and severity of epileptic episodes. Moreover, during the abovementioned stages of the menstrual cycle, in patients with multiple sclerosis, there is an aggravation of physical and cognitive symptoms [70,73]. Overall, this leads us to hypothesise that differences in sex hormone profiles could shape disparities in susceptibility and disease manifestations in neuropsychiatric disorders related to disturbances of Tryp metabolism.

**Table 3 ijms-24-06010-t003:** Highlights of sex hormones in tryptophan metabolism: healthy condition.

Sex Hormone	Study	Effect on Tryptophan Metabolism
Oestrogen	Preclinical	↓ extracellular 5-HT in the hypothalamus [74,75] ↓ hippocampal brain levels of Tryp (ATD condition) [29]
	Clinical	↓ cortical trapping of α-[^11^C]methyl-L-tryptophan ^$^ [76] ↓ circulating levels of plasma Tryp [77] ↓ circulating levels of plasma kyn and picolinic acid [78]
Progesterone	In vitro	↑ KYNA levels (cultures of human macrophages) [79] ↓ QUIN levels (cultures of human macrophages) [79]
	Preclinical	↑ brain levels of KYNA after administration of L-kyn [80] ↓ L-kynurenine-induced cortical spreading depression [80]
	Clinical	↑ circulating levels of plasma kyn and urinary kyn excretion [81] ↑ circulating levels of plasma KYNA [17] ↓ inflammation-induced activation of IDO [82] ↓ Tryp catabolism to kyn and neurotoxic metabolites [78,82]
Testosterone	Preclinical	Lack of testosterone attenuated hippocampal KYNA levels (sleep deprivation condition) [30]
	Clinical	↑ circulating levels of plasma Tryp [77]

5-HT: serotonin; ATD: acute Tryp depletion; kyn: kynurenine; KYNA: kynurenine acid; IDO: indoleamine-2, 3-dioxygenase; QUIN: quinolinic Acid; Tryp: tryptophan; ^$^ radiotracer analogue of tryptophan used for the measurement of brain serotonin synthesis under normal circumstances; ↑: increase; ↓: decrease.

#### 1.3.1. Oestrogen

Oestrogen exerts neuroprotective actions in females by mediating mitochondrial function, anti-apoptotic mechanisms, and immune responses [22]. However, variations of this hormone across the menstrual cycle are also associated with serotonergic vulnerability, which could be an additional risk of mood disorders [19,22]. In agreement, preclinical data showed that oestrogen played a neuromodulatory role in the 5-HTergic system by decreasing the extracellular hypothalamic 5-HT synthesis [74,75]. Furthermore, neuroimaging studies in healthy women scanned immediately before ovulation or in their follicular phase exhibited lower cortical 5-HT production than healthy men using α-[^11^C]methyl-L-tryptophan [76]. Oestrogen release has also been proven to modulate Tryp levels and pro-inflammatory changes [29,77,78]. In fact, ATD resulted in a higher reduction of hippocampal Tryp levels in female rats responding to the oestrogen peak compared with males and females on other hormonal phases of their cycle [29]. Accordingly, clinical evidence indicated that oestrogen reduces circulating levels of plasma Tryp [77], that, in turn, was correlated with pro-inflammatory variations, including increased interleukin 6 (IL-6) and a decrease of kynurenine and picolinic acid [78].

#### 1.3.2. Progesterone

Progesterone is a critical physiological component in women that also mediates Tryp catabolism and the production of essential compounds from the oxidation pathway [82]. In particular, animal research showed that progesterone facilitates the reduction of L-kynurenine-induced cortical spreading depression by increasing brain levels of KYNA [80]. Furthermore, human studies demonstrated that progesterone levels were positively correlated with plasma concentrations of kynurenine [81] and KYNA [17]. Further, it was observed that progesterone attenuated the inflammation-induced activation of indoleamine-2,3-dioxygenase (IDO) and reduced Tryp catabolism to kynurenine-related neurotoxic metabolites [78,82]. An in vitro human macrophage study presented similar results, in which progesterone decreased QUIN levels and increased neuroprotective KYNA concentrations [79].

#### 1.3.3. Testosterone

Testosterone is a pivotal sex hormone that exerts neuroprotective actions, mostly in men. In fact, the higher susceptibility of boys to neurodevelopmental disorders such as ASD and ADHD had been postulated to be a result of abnormal testosterone levels during the development of the male brain [22]. Similar to oestrogen and progesterone, testosterone has potential implications for brain function, including 5-HT synthesis. Indeed, low doses of this hormone have been proposed to increase brain levels of 5-HT and 5-Hydroxyindoleacetic acid (5-HIAA), while high doses have been presented to have the opposite effect [52]. However, few studies have focused on studying the impact of testosterone on Tryp metabolism. According to a previous hypothesis by Badawy et al. [52], one clinical study showed that testosterone treatment increases circulating plasma levels of Tryp [77]. Further, a preclinical study showed that gonadectomy in male animals (that lacked circulating testosterone) attenuated an elevation in hippocampal KYNA levels after sleep deprivation [30].

## 2. Results and Discussion

This research only included original English-written articles from the past 20 years. The initial search identified 2172 references, of which 199 were from Pubmed, 1115 from Scopus, and 858 from Web of Science. After removing the duplicates, 1463 references were revised by title and abstracts considering the inclusion and exclusion criteria previous described. After that first revision, 40 articles were selected for a full reading. This literature review spotlighted only clinical studies covering sex differences and Tryp metabolism for neuropsychiatric conditions. According to that purpose, 17 references were eligible for analysis and data extraction (Figure 2). The overall summary of those studies is described in Table 4. In the following sections, the link between biological sex and Tryp metabolism and their possible impact on neuropsychiatric disorders will be addressed.

### 2.1. Sex Differences in Tryptophan Metabolism: Neuropsychiatric Conditions

Neuropsychiatric conditions are very heterogenous, with different risk factors and biological dysfunctions that lead to a spectrum of symptoms. Consequently, several molecular and cellular mechanisms may be involved in disease expression and progression. Changes in Tryp metabolism have been consistently associated with the pathogenesis of various neuropsychiatric diseases [1,9,10,14,18,19,20]. Specifically, alterations in 5-HTergic function have been observed in epilepsy and psychiatric conditions such as depression, ADHD, ASD, and OCD [19,27,97,98]. In contrast, changes in the kynurenine pathway have been established as key players in inflammation-associated disorders, as well as for epilepsy, and neurodegenerative and autoimmune disorders [10,14,20,99]. Since neuroinflammation is a hallmark widely related to neurological and psychiatric diseases [11], many studies explored the possible role of the Tryp oxidation pathway [57,100,101]. Indeed, Tryp-kynurenine pathway activation seems relevant to the pathophysiology of neurological abnormalities by activating the neurotoxic branch and suppressing the neuroprotective one [53,102,103,104,105,106,107]. As a result, there is a shift towards the Tryp oxidation pathway, depleting 5-HT and increasing kynurenine neurotoxic metabolites [9,108,109]. Further, several authors also reported a sex-dependent molecular underpinning of Tryp metabolism in some neuropsychiatric disorders (Table 5).

#### 2.1.1. Sexual Dimorphisms on Tryptophan Metabolism: Female Sex Bias for Neuropsychiatric Prevalence

Both serotonergic dysfunction and systemic inflammation were associated with depression [10,110,111] and AD [10,58,103], while only neuroinflammation has been postulated as a critical element in neuropsychological symptoms of chronic fatigue syndrome (CFS) [100,109]. Further, compared with healthy individuals, AD and depressive patients had reduced circulating levels of Tryp [58,103,110,111], while CFS patients exhibited similar levels [112,113]. Concerning sexual dimorphisms, women with MDD and AD showed a reduced availability of Tryp compared with men [45,83,94,95]. Further, depressive women also had lower 5-HT production [45] and a more pronounced depressive response to the depletion of this neurotransmitter precursor [85,86]. In contrast, for CFS both sexes had similar concentrations of this amino acid available in circulation for baseline conditions [42,87] and following a supplementation treatment [87]. Regarding kynurenine, some studies reported higher levels in circulation in women with AD and CFS [42,95], while others reported no sex differences in that metabolite for CFS and depression [42,87]. All these results sustained that Tryp signalling might be involved in the sexual dimorphism observed in the pathology and symptomatology of depression and AD. However, Tryp metabolism may not be related to sexual dimorphisms in CFS, which is consistent with the fact that this molecular underprint is also not markedly altered for this neurological disorder [112,113]. Interestingly, PET studies using α-[^11^C]methyl-L-tryptophan appear to show different results from cerebrospinal fluid analysis for MDD. On the one hand, women had reduced levels of free Tryp compared with men [45]. On the other, PET data showed increased uptake of α-[^11^C]methyl-L-tryptophan in multiple regions of the prefrontal cortex and limbic system for women [88]. Although usually α-[^11^C]methyl-L-tryptophan is used as an approximation of 5-HT brain synthesis rate, this tracer uptake may also indicate changes in the kynurenine pathway [114]. So, it is important to have caution with the use of this tracer to measure serotonergic signalling specifically for pathological conditions.

Overall, due to the reported reduction of Tryp availability and 5-HT synthesis in the female sex, it is possible to postulate that these alterations may be implicated in the female sex bias in depression. Furthermore, sexual dimorphism in the progression of AD could be related to disturbances in the oxidation pathway. Indeed, it was reported that neurodegeneration biomarkers, such as amyloid-β and neurofilament light chain, were positively correlated with the plasma kynurenine/Tryp ratio [115] and that for this condition, women exhibited higher plasma kynurenine levels than men [95]. Similarly, as neuroinflammation is also involved in the pathophysiology of MDD [116], increased accumulation of α-[^11^C]methyl-L-tryptophan in women can reflect an activation of the oxidative pathway rather than of the hydroxylation pathway [88]. However, this radiotracer is not specific to the 5-HT or kynurenine pathway, so further studies are needed to validate this assumption.

#### 2.1.2. Sexual Dimorphisms on Tryptophan Metabolism: Non-Female Sex Bias for Neuropsychiatric Prevalence

Neuropsychiatric disorders could be divided into their distinct prevalence between sexes. For instance, MDD, AD, and CFS are more prevalent in women, while substance/drug abuse disorders, ASD, and ADHD in men [24,117,118,119]. Interestingly, in neuropsychiatric disorders in which women are not the susceptible sex, there is less evidence of sexual dimorphisms involving Tryp metabolism. Further, we also found contradictory data regarding serotoninergic alterations for addiction. On one hand, some studies reported similar plasma levels of Tryp and 5-HT for both sexes [44,91]. On the other hand, Pivac et al. observed lower platelet 5-HT levels in women with addiction problems [84]. Moreover, women polydrug ecstasy users following ATD showed a more significant mood lowering relative to their controls than men [90]. On the contrary, among methamphetamine users, only males exhibited downregulated plasma Tryp levels compared with controls [96]. Since individuals with addiction problems have high comorbidity with other psychiatric problems, this may explain the different results mentioned above [44,84]. For ADHD, although boys tend to have more severe manifestations, girls had increased susceptibility following ATD. Namely, it was observed that these girls had increases in aggressiveness after a low level of provocation [89].

Changes in kynurenine metabolites were reported in bipolar and addictive women, which presented reduced levels of plasma KYNA compared with men [44,92]. Even though women are not the predisposed sex for both conditions, they experience more severe symptoms related to these disorders [117,118,119]. It is thus plausible that the reduced capacity of women to synthesise KYNA could be implicated. Accordingly, following ATD, women with polydrug ecstasy consumption exhibited increased uptake of α-[^11^C]methyl-L-tryptophan in frontal regions compared to men [91], which may suggest a higher kynurenine pathway activation. However, once again, a higher serotonin synthesis cannot be excluded, as this radiotracer is not specific for any particular pathway.

#### 2.1.3. Current Understanding of How Sexual Dimorphisms on Tryptophan Metabolism May Be Involved in the Onset and Progression of Neuropsychiatric Diseases

Sex-specific changes in Tryp metabolism of neurological/psychiatric illnesses have only been explored to a minimal extent in the literature. Moreover, there is still no direct link between sex differences in Tryp metabolism and the onset and progression of neuropsychiatric diseases. Although, some sexual dimorphisms in Tryp metabolism for some neuropsychiatric conditions are consistent with the dysfunction of this amino acid metabolism and its relation to disease severity (Table 6).

In depression and related disorders, sexual dimorphisms on Tryp metabolism reveal that women, compared with men, had a reduced Tryp pool and increased vulnerability to transient depressive symptoms after Tryp depletion [45,83,85,86,94]. Considering that there is a female-sex bias in the development and progression of those conditions and that those patients exhibit the same dysfunctions compared with healthy controls [1,9,10,19,25,26,110,111,116], a link between the sexual dimorphisms on Tryp dysfunction and disease onset and progression may be present. Furthermore, the same hypothesis can be made in AD, in which women are also more vulnerable than men. Here, sexual dimorphisms on Tryp metabolism reveal that women had a reduced Tryp pool and increased circulating kynurenine levels compared with men [95], which are the same disturbances reported in AD patients as associated with neurodegeneration progression [9,10,58,103,115].

On the other hand, for disorders that affect more men than women, sexual dimorphisms on Tryp metabolism and its possible relationship with pathological onset and progression are not so clear. On one hand, for ASD and schizophrenia, sexual dimorphisms on Tryp metabolism reveal that men, compared with women, have increased production of 5-HT [126,127] and KYNA [122], respectively. Similarly, patients with those conditions also exhibit the same dysfunctions compared to healthy controls [9,10,15]. Moreover, specifically for ASD, the inverse correlation of 5-HT with the score for intellectual ability was more significant in males [127], suggesting that sexual dimorphisms on Tryp metabolism could be related to the sex bias in those disorders. On the other hand, for some conditions more prevalent in men (e.g., substance/drug abuse disorder and ADHD), sexual dimorphisms suggest that women are more vulnerable to some dysfunctions related to Tryp metabolism [44,89]. Particularly, in substance/drug abuse disorder, women, compared with men, have increased activation of the kynurenine pathway with a reduced capacity to synthesise KYNA [44], and for ADHD, increased susceptibility to developing aggressive behaviour following Tryp depletion [89]. We believe that those results could be related to comorbidity of other psychiatric problems in substance/drug abuse disorder and that a higher susceptibility of women to suffer serotoninergic alterations due to changes in the levels of its precursor, previously reported also for healthy conditions, could be involved in the results in ADHD.

Finally, for most neuropsychiatric disorders with an almost equal sex ratio on prevalence (e.g., HD, OCD, and epilepsy), no data on sexual dimorphism for Tryp metabolism is available. The exception is bipolar disorder, where sexual dimorphisms arrive, indicating that women had a reduced Tryp pool and KYNA production [92], and men increased circulating kynurenine levels [121]. Interestingly, both of those Tryp dysfunctions are associated with the pathological condition of that disorder [120,121].

From all those results, female-sex bias in neuropsychiatric diseases appears to be involved in the reduced availability of the Tryp pool and depletion of 5-HT synthesis. It is also plausible to hypothesise that women may have a reduced capacity to synthesise KYNA in neuropsychiatric conditions compared with men. In contrast, male-sex bias may be implicated in the excessive production of 5-HT and KYNA.

#### 2.1.4. Limitations and Future Perspectives

The study of sexual dimorphism should include variables that highly differ between women and men and can shape Tryp metabolism (Section 1.3). So, it is essential to highlight some crucial differences and limitations between studies, which may have originated the observed discrepancies between them. First, the sample size and the statistical power are very distinct between studies ranging between N = 16 and N = 695. In some cases, the patient’s medication was not an exclusion criteria [44,45,85,94,95], as opposed to others, in which patients were medication-free [84,87,88,90,91,93]. Similarly, the presence of other comorbidities was a factor of exclusion only in some studies [83,85,87,88,89,90,91,92,93]. All those variables may have influenced the results, causing bias. Another factor that can impact the clinical studies is the sex hormone profile. Most of the studies in this review did not specify the menstrual phase women were in during experimental analysis. Further, the studies that did, only included women in the follicular phase or post-menopausal taking hormone replacement therapy [45,87,88,90,91]. Even though that choice is not surprising as it is more convenient to observe women when hormonal levels are at their lowest (follicular phase), that is not a good representative dataset for females. Overall, the involvement of the female menstrual cycle phase in alterations of Tryp metabolism and brain 5-HT for neurological conditions has not been adequately addressed. Future studies should include women at different stages of their menstrual cycle to overcome this gap.

The eating behaviour of participants is also an important variable since Tryp is obtained exclusively from the diet [2,49]. Considering that the half-life of tryptophan is short, around 2 h [128], overnight fasting should be enough to correctly evaluate Tryp supplementation and/or depletion. Nevertheless, analysis of plasma Tryp metabolites could be a good approach to examine the effects on daily life conditions (without fasting). Additionally, the use of animal models could be an advantage to study the impact of Tryp availability as their diet is easily controlled by researchers.

Finally, neuroimaging studies with α-[^11^C]methyl-L-tryptophan, which has been widely used as a serotoninergic marker, must be interpreted taking into account the kynurenine pathway. We also advert that this radiotracer should only be used to estimate brain 5-HT synthesis for normal circumstances and not in pathological states.

## 3. Methods

The methods of this systematic review followed the PRISMA (Preferred Reported Items for Systematic Review and Meta-Analysis) guidelines [129].

### 3.1. Study Eligibility

The inclusion criteria were: studies analysing at least one of the tryptophan metabolites; studies including human subjects with neurological and/or psychiatric conditions; and studies including both biological sexes (female and male). On the other hand, the exclusion criteria were: 1—studies on non-human subjects; 2—studies only including one of the two biological sexes (female or male); case reports, literature reviews, systematic review, and meta-analyses; studies not published in English; and studies older than 20 years.

### 3.2. Search Strategy

This study systematically revised sex differences in Tryp metabolism, focusing on neuropsychiatric diseases. The last research was conducted on 18 January 2023 and included the results of the last 20 years. A search of the literature was performed in Pubmed, Scopus, and Web of Science databases, using the search formulas presented in Table 7. We only included studies conducted on humans, excluded meta-analysis, reviews, and systematic reviews and restricted our search to English-language papers. Only the studies focused on sex differences in neurological conditions were selected.

## 4. Conclusions

Notable findings can be garnered from this review. One of the most important is that Tryp metabolism, the common denominator linking 5-HT dysfunctions and neuroinflammation, is a preeminent aspect involved in several neuropsychiatric conditions. Consistent evidence suggests that women have higher susceptibility than men to suffer serotoninergic alterations due to changes in the levels of its precursor Tryp. Indeed, female sex bias in neuropsychiatric diseases is involved in a reduced availability of this amino acid pool and 5-HT synthesis (Figure 3). In addition, the sexual dimorphisms found in this specific molecular mechanism correlate with a distinct predisposition between women and men for some neuropsychiatric disorders. Perhaps the most striking message of this review is the relevance of deepening the study of Tryp metabolism behind sex differences for neuropsychiatric disorders. Despite the well-established understanding that women and men differ in their predisposition and disease manifestations for neuropsychiatric conditions, sex is rarely considered when making diagnostic or treatment decisions. In fact, sex-specific changes in Tryp metabolism of neurological/psychiatric illnesses have only been explored to a minimal extent in the literature. Further, worryingly, the effect of diet and sex steroids, both involved in this molecular mechanism, have been poorly addressed. We believe that a better understanding of the Tryp metabolic molecular processes behind these sex differences could help develop more targeted therapies with higher success rates, especially in diseases where sex differences are most prominent.

## Figures and Tables

**Figure 1 ijms-24-06010-f001:**
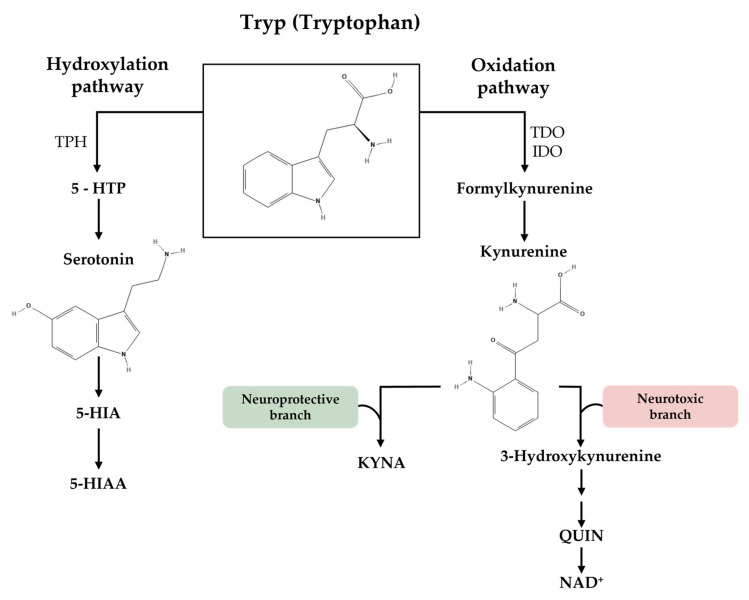
Tryptophan as a precursor of several biologically essential compounds. The metabolism of Tryp leads to the generation of several neuroactive compounds within the CNS. These include the neurotransmitter 5-HT and products of the kynurenine pathway. In the 5-HT pathway (left), Tryp is catalysed by TPH to produce 5-HTP, precursor of 5-HT, which is further metabolised to 5-HIA, followed by oxidation to 5-HIAA. In the kynurenine pathway (right), TDO and IDO catalyse the synthesis of kynurenine giving rise to critical neuroactive compounds such as QUIN and KYNA, often termed the “neurotoxic” and “neuroprotective” branches, respectively. 5-HIA—5-hydroxindole acetaldehyde; 5-HIAA—5-hydroxyindole acetic acid; 5-HT—serotonin; 5-HTP—5-hydroxytryptophan; CNS—central nervous system; IDO—indoleamine 2,3-dioxygenase; KYNA—kynurenic acid; NAD+—nicotinamide adenine dinucleotide; QUIN—quinolinic acid; TDO—tryptophan 2,3-dioxygenase; TPH—tryptophan hydroxylase; Tryp—tryptophan. The Figure was partly generated using PubChem for the chemical structure of tryptophan (, accessed on 27 January 2023), kynurenine (https://pubchem.ncbi.nlm.nih.gov/compound/846#section=2D-Structure, accessed on 27 January 2023), and serotonin (https://pubchem.ncbi.nlm.nih.gov/compound/5202#section=2D-Structure, accessed on 27 January 2023).

**Figure 2 ijms-24-06010-f002:**
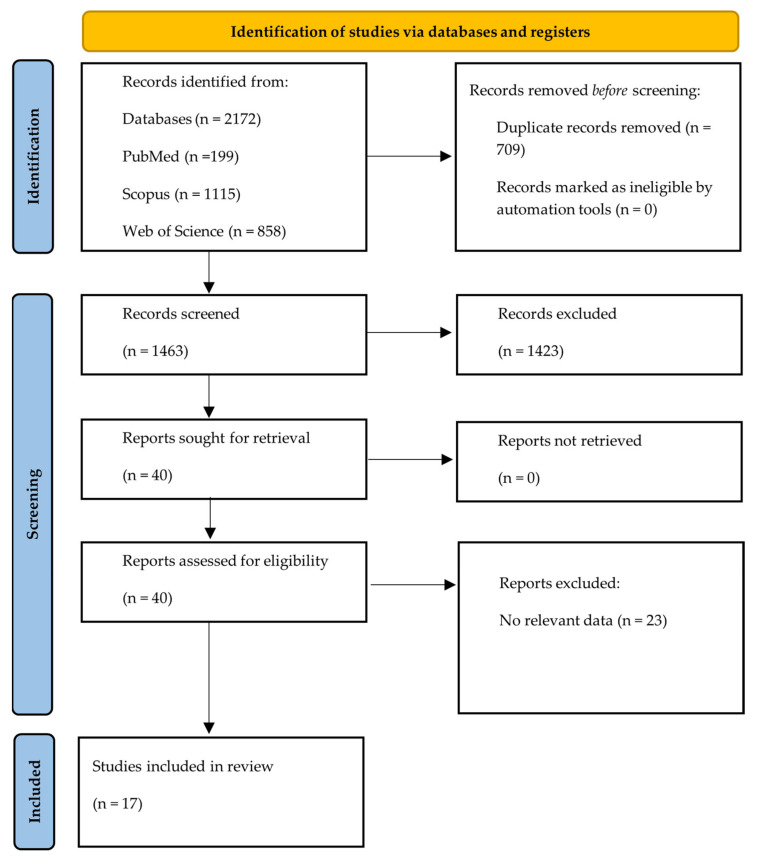
PRISMA flow diagram of the study research methodology in the literature. This literature review spotlighted only clinical studies focusing on sex differences and Tryp metabolism for neurological conditions and excluded reviews, meta-analyses, and systematic reviews or unavailable as complete articles. Further, this research only included original English-written articles from the past 20 years. Articles referred as “no relevant data” did not allow us to draw conclusions regarding sexual dimorphisms (the two biological sexes were not divided as independent groups and/or a direct statistical comparison between sexes was not provided).

**Figure 3 ijms-24-06010-f003:**
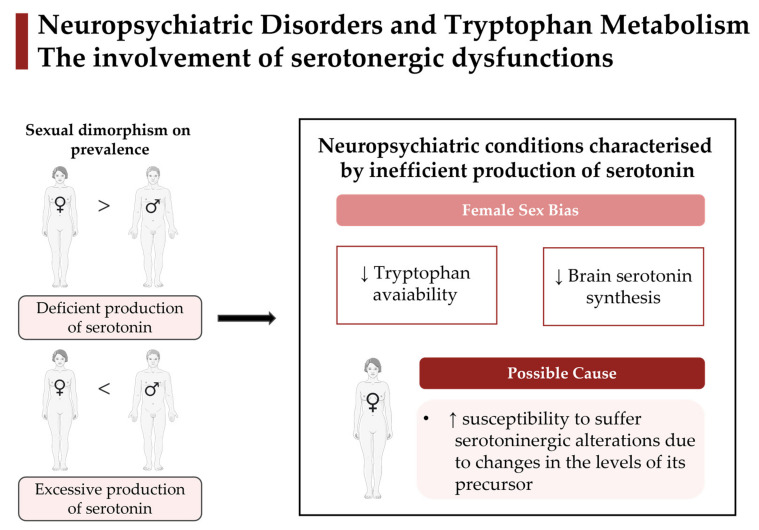
Sexual bias for some neuropsychiatric conditions seems to be related to opposite disturbances in the (5-HT) system. On one hand, most neuropsychiatric disorders with aggravated manifestations in women (e.g., depression, affective, and mood disorders) are associated with deficient production of serotonin. On the other hand, some conditions that affect more males than females (e.g., ASD and ADHD) have excessive production of that neurotransmitter. Specifically, female sex bias in neuropsychiatric diseases appears to be involved in a reduced availability of this amino acid pool and 5-HT synthesis. Further, women have higher susceptibility than men to suffer serotoninergic alterations due to changes in the levels of its precursor Tryp. These changes in Tryp metabolism and, consequently, in the serotoninergic system can lead to sexual dimorphisms on the prevalence and severity of neuropsychiatric disorders. The Figure was partly generated using Servier Medical Art, provided by Servier, licensed under a Creative Commons Attribution 3.0 unported license.

**Table 1 ijms-24-06010-t001:** Highlights of differences in tryptophan metabolism between sexes: healthy conditions.

	Path	Study	Sexual Dimorphisms (♀ vs. ♂)
Peripherical levels	Tryp	Preclinical	↑ plasma Tryp levels [29,30,31]
		Clinical	↑ plasma-free Tryp levels [5,39] (=) plasma Tryp levels [42,43] ↓ plasma Tryp levels [44,45]
	5-HT	Preclinical	↑ plasma 5-HT/Tryp ratio [32,33]
		Clinical	-
	Kyn	Preclinical	↓ plasma Kyn levels [30] (=) plasma KYNA levels [30] (=) plasma Kyn/Tryp ratio [33]
		Clinical	↓ KYNA plasma levels [17] (=) Kyn and KYNA plasma levels [39]
Brain levels	Tryp	Preclinical	↑ brain Tryp levels (brainstem, striatum and cortex) [35] ↑ brain Tryp levels (whole brain) [34]
		Clinical	↑ plasma Tryp/CAA ratios [5,39] (=) plasma Tryp/CAA ratios [42]
	5-HT	Preclinical	↑ 5-HT levels (brainstem and limbic forebrain) and 5-HIAA (whole brain) [35] ↑ 5-HIAA/5-HT ratios (hypothalamus/preoptic area and limbic forebrain) [35] ↑ 5-HIAA/5-HT ratios (brainstem—dorsal raphe nucleus) [36] ↑ 5-HT and 5-HIAA (limbic forebrain—hippocampus) [37]
		Clinical	↑ cerebrospinal fluid 5-HIAA levels [40,41]
	Kyn	Preclinical	(=) KYNA levels (cortex and hippocampus) [30] (=) Kyn/Tryp ratios (whole brain) [33]
		Clinical	-

5-HIAA: 5-hydroxy indole acetic acid; 5-HT: serotonin; CAA: competing amino acids; Kyn: kynurenine; KYNA: kynurenine acid; Tryp: tryptophan; Vs.: versus/opposing; ♀: women/females; ♂: men/males; ↑: increase; ↓: decrease; (=): similar.

**Table 2 ijms-24-06010-t002:** Highlights of the impact in tryptophan nutritional status between sexes: healthy condition.

Condition	Parameters	Sexual Dimorphisms (♀ vs ♂)
Tryp supplementation	Biochemical	↑ plasma-free Tryp [39] (=) plasma total kynurenines [39]
	Behavioural	(≠) interaction between Tryp and mood (a positive trend in men and a negative trend in women) [5]
Tryp depletion	Biochemical	↓ trapping of α-[^11^C]methyl-L-tryptophan ^$^ (whole brain) [59]
	Behavioural	↑ memory impairment [60] * ↑ mood-lowering [61] ↑ mood-lowering with pre-existing aggressive traits [62] ↑ impairment of the recognition of fearful facial expressions [63]

Tryp: tryptophan; vs: versus/opposing; * women’s menstrual cycle was determined and considered in the studies of this review; ^$^ radiotracer analogue of tryptophan used for the measurement of brain serotonin synthesis under normal circumstances; ♀: women/females; ♂: men/males; ↑: increase; ↓: decrease; (=): similar; (≠): different.

**Table 4 ijms-24-06010-t004:** Summary of clinical studies included in this review.

Reference	Sample Size	Health/Condition	Assessment Methods	Major Findings Between Sexes (♀ vs. ♂)
Bano et al. (2004) [83]	N= 16 (Did not specify the sample size of each sex) Randomised controlled clinical trial	Depressed patients treated with Prozac ^#^ for four weeks	Fluorometric method on blood samples	↓ in total plasma Tryp concentrations relative to controls (56% on ♀; 33% on ♂ - before treatment). ↑ in total plasma Tryp concentrations relative to controls (83% on ♀; 32% on ♂—after four weeks of treatment).
Pivac et al. (2004) [84]	Healthy controls: N = 233 (123 women and 110 men) Alcohol dependency patients: N = 190 (42 women and 148 men)	Drug-free patients with AUD	Fluorimetric method on platelet-rich-blood samples	↓ platelet 5-HT concentrations.
Badawy et al. (2005) [42]	Healthy controls: N= 42 (16 females and 26 males) CFS patients: N= 23 (14 females and 9 males)	CFS patients	Fluorimetric method on blood samples	↑ levels of plasma Kyn. (=) levels of plasma Tryp.
Booij et al. (2005) [85]	N = 21 (10 women and 11 men) Double-blind cross-over approach	Remitted depression participant’s in ATD condition	Psychiatric symptoms and behavioural measures of depression	↑ depressive response (ATD condition).
Moreno et al. (2006) [86]	N = 59 (41 women and 18 men)	Remitted depressive patients in ATD condition	Behavioural measures of depressive symptoms	↑ depressive response (ATD condition).
Weaver et al. (2010) [87]	Healthy controls: N= 16 (10 women * and 6 men) CFS patients: N= 22 (15 women* and 7 men)	CFS participants in Tryp enhancement condition	HPLC on blood samples	(=) levels of plasma Tryp and Kyn (Tryp supplementation condition). Upregulated plasma prolactin ^&^ responses relative to controls (only on ♀)
Frey et al. (2010) [88]	N= 25 (13 women * and 12 men)	MDD patients	PET with α-Methyl-L-tryptophan ^$^	↑ uptake tracer in multiple regions of the prefrontal cortex and limbic system.
Kötting et al. (2013) [89]	N = 20 (10 girls and 10 boys) Double-blind within-subject cross-over approach	ADHD participants in control or ATD condition	Behavioural measures of aggression 2.5 h after administration	(≠) response under ATD (↑ aggressively after aggressive lower provocation on ♀; ↑ aggressively after aggressive high provocation on ♂).
Young et al. (2014) [90]	Ecstasy user participants: N = 13 (7 women * and 6 men) Ecstasy non-user participants: N = 17 (9 women* and 8 men)	Participants who had used ecstasy at least 25 times in control or ATD condition	Behavioural measures of mood states and impulsivity	(≠) mood reduction relative to controls (↓ only on ♀) (ATD condition).
Booij et al. (2014) [91]	Healthy controls: N= 18 (9 women * and 9 men) Ecstasy polydrug users: N = 17 (8 women * and 9 men)	Ecstasy polydrug users	HPLC on blood samples PET with α-Methyl-L-tryptophan ^$^	(=) levels of plasma Tryp. ↓ uptake tracer in the frontal regions relative to controls (especially noticeable in men).
Platzer et al. (2017) [92]	Healthy participants: N = 93(57 women and 37 men) Bipolar patients: N =68 (26 women and 42 men)	Bipolar patients	UPLC on blood samples	↓ levels of plasma Kyn and KYNA.
Hestad et al. (2017) [45]	Healthy participants: N = 31(22 women and 9 men) Depression patients: N =44 (23 women and 21 men)	Depressed patients	HPLC on blood and cerebrospinal fluid	↓ levels of Tryp in both plasma and cerebrospinal fluid. (=) levels of Kyn in both plasma and cerebrospinal fluid.
Zhou et al. (2018) [93]	Healthy participants: N = 72(31 women and 41 men) MDD patients: N =146 (70 women and 76 men)	MDD patients	HPLC-MS/MS on blood samples Behavioural measures of depressive symptoms, cognition, and memory	Negative association between Kyn levels and behavioural measures of cognition (only on ♀)
Vidal et al. (2020) [44]	Healthy controls: N= 80 (40 women and 40 men) AUD patients: N= 130 (35 women and 95 men)	AUD patients with a high prevalence of psychiatric comorbidity	HPLC on blood samples	(≠) levels of plasma Tryp relative to controls (↑ on ♀; ↓ on ♂). ↑ levels of plasma Kyn. ↓ levels of plasma 5-HT and KYNA. (≠) levels of plasma Tryp in participants with comorbid substance use and mental disorders relative to non-comorbid (↑ on ♀; ↓ on ♂).
Reininghaus et al. (2019) [94]	N= 426 (242 women and 184 men)	MDD patients over a 6-week rehabilitation treatment	HPLC on blood samples	↓ levels of plasma Kyn and Kyn/Tryp ratio (after treatment).
Xu et al. (2021) [95]	N = 695 participants (377 women and 318 men)	AD patients	Metabolomics analysis over 665 plasma metabolites on blood samples	↓ levels of plasma kynurenate (Kyn pathway) relative to controls (only on ♂). ↓ levels of plasma Tryp betaine (Tryp pathway) relative to controls (only on ♀).
Cheng et al. (2023) [96]	N = 27 (10 women and 17 men)	Methamphetamine addicts	UHPLC–MS/MS on blood samples	L-tryptophan was downregulated relative to controls (only on ♂).

5-HIAA: 5-Hydroxy indole acetic acid; 5-HT: serotonin; AD: Alzheimer’s disease; ADHD: attention deficit hyperactivity disorder; ATD: acute tryptophan depletion; AUD: alcohol use disorders; CFS: chronic fatigue syndrome; HPLC: high-performance liquid chromatography; HPLC-MS/MS: high-performance liquid chromatography with tandem mass spectrometry; Kyn: kynurenine; KYNA: kynurenine acid; MDD: major depressive disorder; OCD: obsessive-compulsive disorder; PET: positron emission tomography; Tryp: tryptophan; UPLC: ultra performance liquid chromatography; * women;s menstrual cycle was determined and considered for this study; ^#^ selective serotonin reuptake inhibitor and it works by blocking the absorption of that neurotransmitter in the brain; ^$^ radiotracer analogue of tryptophan used for the measurement of brain serotonin synthesis under normal circumstances; ^&^ serotonergic-based responses; ♀: women; ♂: men; ↑: increase; ↓: decrease; (=): similar; (≠): different.

**Table 5 ijms-24-06010-t005:** Highlights of differences in tryptophan metabolism between sexes: neuropsychiatric conditions.

	Path	Condition	Sexual Dimorphisms (♀ vs. ♂)
Peripherical levels	Tryp	MDD	↓ plasma Tryp levels [45,83] ↓ plasma Tryp levels [94]
		CFS	(=) plasma Tryp levels [42] (=) plasma Tryp levels [87] * (=) plasma Tryp and Kyn levels (Tryp supplementation condition) [87] *
		Substance/drug abuse	(=) plasma Tryp levels [44,91] (≠) shift on plasma Tryp levels relative to controls (↓ on ♂; (=) on ♀) [96]
		AD	↓ plasma Tryp levels [95]
	5-HT	Substance/drug abuse	↓ platelet 5-HT levels [84] (=) plasma 5-HT levels [44]
	Kyn	CFS	↑ plasma Kyn levels [42] (=) plasma Kyn levels [87] *
		Bipolar disorder	↓ plasma Kyn and KYNA levels [92]
		MDD	(=) plasma Kyn levels [45]
		Substance/drug abuse	↑ plasma Kyn levels [44] ↓plasma KYNA levels [44]
		AD	↑ plasma Kyn levels [95]
Brain levels	Tryp	MDD	↓ cerebrospinal fluid Tryp levels [45] ↑ depressive response (ATD condition) [85,86] ↑uptake tracer of α-Methyl-L-tryptophan ^$^ (prefrontal cortex and limbic system) [88] *
		ADHD	↑ relative risk of presenting with increased aggression after a low level of provocation (ATD condition) [89]
		Substance/drug abuse	↑uptake tracer of α-Methyl-L-tryptophan^$^ (frontal regions) [91] * (≠) mood reduction relative to controls (↓ only on ♀) (ATD condition) [90] *

5-HIAA: 5-hydroxy indole acetic acid; 5-HT: serotonin; AD: Alzheimer’s diseases; Kyn: kynurenine; KYNA: kynurenine acid; CFS: chronic fatigue syndrome; MDD: major depressive disorder; OCD: obsessive-compulsive disorder; * women’s menstrual cycle was determined and considered for this study; **^$^** radiotracer analogue of Tryp used for the measurement of 5-HT synthesis under normal circumstances; Tryp: tryptophan; Vs.: versus/opposing; ♀: women; ♂: men; ↑: increase; ↓: decrease; (=): similar; (≠): different.

**Table 6 ijms-24-06010-t006:** Summary of the reported dysfunctions of tryptophan metabolism, its relation to disease severity, and described sexual dimorphisms for some neuropsychiatric conditions.

Condition	Sex-Bias on Incidence (♀:♂)	Dysfunctions of Tryp Metabolism	Tryp metabolism and Disease Severity	Sexual Dimorphisms and Dysfunctions of Tryp Metabolism	Reference
CFS	4:1	No reported dysfunctions	-	No reported sexual dimorphisms	[44,48,112,113]
Depression and related disorders	2:1	↓Circulating Tryp ↓Brain 5-HT and 5-HIAA ↑Circulating Kyn IDO ↑Brain QUIN	Tryp depletion and lowering of brain 5-HT were associated with transient depressive symptoms. Elevated Kyn, IDO and QUIN levels were associated with depressive symptoms.	↓Circulating Tryp (♀ vs. ♂) ↑Depressive response to Tryp depletion (♀ vs. ♂)	[1,9,10,19,25,26,45,83,85,86,94,110,111,116]
AD	2:1	↓Circulating Tryp and 5-HT ↑Circulating Kyn ↑Brain kynurenine, IDO and QUIN	Neurodegeneration biomarkers, such as amyloid-β, senile plaques and neurofilament light chain, were positively correlated with the plasma Kyn/Tryp ratio and IDO levels.	↓Circulating Tryp (♀ vs. ♂) ↑Circulating Kyn (♀ vs. ♂)	[9,10,58,95,103,115]
HD	1:1	↑Brain Kyn and QUIN	Symptom severity was associated with plasma Kyn/Tryp ratio. QUIN induced behavioural and motor deficits in Huntingtin models.	No data	[18,20,92]
Bipolar Disorder	1:1	↓Circulating Tryp and KYNA ↑Circulating Kyn	Plasma Kyn/Tryp ratio was associated with depressive severity while trending toward a negative association with mania symptoms.	↓Circulating Tryp and KYNA (♀ vs. ♂) ↑Circulating Kyn (♂ vs. ♀)	[120,121]
OCD	1:1	↑Brain α-[^11^C]methyl -L-tryptophan^$^ uptake	Symptom severity correlated positively with uptake from the caudate and temporal lobe.	No data	[97]
Epilepsy	1:1	↑Brain α-[^11^C]methyl -L-tryptophan^$^ uptake, serotonin immunoreactivity, 5-HIAA and QUIN ↑Circulating 3HK and QUIN	Brain α-[^11^C]methyl-L-tryptophan ^$^ uptake was correlated with seizure onset.	No data	[10,98]
Schizophrenia	1:1.4	↑Brain and cerebrospinal fluid KYNA	Increased KYNA levels were correlated with cognitive impairments.	↑Circulating KYNA (♂ vs. ♀)	[9,10,122]
Substance/ drug abuse disorder	1:2	↑Circulating Kyn	Reduced Tryp availability and deficient 5-HT synthesis as higher serum levels of Kyn had been associated with increased consumption and vulnerability to addiction.	↑Circulating Kyn (♀ vs. ♂) ↓Circulating KYNA (♀ vs. ♂)Incoherent results on 5-HT	[44,84,96,98,123,124]
ADHD	1:2.3	↓ 3HK ↑Circulating Kyn, KYNA and 5-HT	Kyn levels were positively correlated to the pathological conditions in a mouse model of ADHD.	↑Relative risk for aggressive behaviour during Tryp depletion (♀ vs. ♂)	[10,27,89,99,125]
ASD	1:3	↓Circulating KYNA ↑Circulating 5-HT, Kyn and QUIN	Decreased central 5-HT associates with increased aggression. Tryp depletion exacerbates ASD symptoms, including stereotypies and repetitive behaviours.	↑Circulating 5-HT (♂ vs. ♀) Platelet 5-HT content was inversely correlated to the score for intellectual ability (♂ vs. ♀)	[15,126,127]
PD	1:3.5	↓Circulating KYNA ↑Brain IDO and QUIN	Kyn inhibitors significantly reduce the severity of dystonia in animal models.	No data	[10,14]

3HK: 3-hydroxykynurenine; 5-HIAA: 5-hydroxy indole acetic acid; 5-HT: serotonin; AD: Alzheimer’s disease; ADHD: attention deficit hyperactivity disorder; ASD: autism spectrum disorder; CFS: chronic fatigue syndrome; HD: Huntington’s disease; IDO: indoleamine 2,3-dioxygenase; Kyn: kynurenine; KYNA: kynurenine acid; OCD: obsessive-compulsive disorder; PD: Parkinson’s disease; QUIN: quinolinic acid; Tryp: tryptophan; ^$^ radiotracer analogue of tryptophan used for the measurement of brain serotonin synthesis under normal circumstances; ♀: women; ♂: men; ↑: increase; ↓: decrease.

**Table 7 ijms-24-06010-t007:** Search strategy used.

Database	Search Formula
Medline (via PubMed)	((((“tryptophan” [Mesh] AND “Sex Characteristics” [Mesh]) OR (“tryptophan” [title] AND “gender” [All fields]) OR (“tryptophan” [title] AND “sex” [All fields])) NOT (“review” [Publication Type])) NOT (“systematic review” [Publication Type])) NOT (“meta analysis” [Publication Type])
Scopus	TITLE-ABS-KEY ((“tryptophan” AND “Sex Characteristics”) OR (“tryptophan” AND “gender”) OR (“tryptophan” AND “sex”)) AND (LIMIT-TO(DOCTYPE, “ar”))
Web of Science Medline	(TS= (“tryptophan” AND (“Sex Characteristics” OR “gender” OR “sex”)))

## Data Availability

Not applicable.

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
