# Peer review of "Sex Differences in Tryptophan Metabolism: A Systematic Review Focused on Neuropsychiatric Disorders"

_ijms, 2023, doi:10.3390/ijms24066010_

Round 1
Reviewer 1 Report
I would like to start by congratulating the authors on their work. The topic is most interesting, the paper is well written and pleasant to read. Notwithstanding, I have a couple of concerns detailed above:
Overall comments
- The title is misleading. I may imagine the authors original intent was to perform a review on Tryp role on neurological disorders, but most of the retrieved studies were on psychiatric conditions. It should be rephrased
- In line with the previous comment, mood disorders, anxiety, ADHD, AUD, are psychiatric conditions. This is not clear over the manuscript and it should be clearly differentiated whenever the authors are discussing neurological disorders (as dementia, epilepsy…) or psychiatric disorders (as the remaining discussed on the article)
- Chronic fatigue syndrome is not a neurological condition and I’m not sure it is considered a psychiatric disorder.
Methods:
- Could the authors provide the reasons for excluding the 1423 articles, by categories?
- What do the authors mean by “no relevant data” for excluding 23 articles? What were the criteria used?
Author Response
Reply to Reviewer #1
1 - The title is misleading. I may imagine the authors original intent was to perform a review on Tryp role on neurological disorders, but most of the retrieved studies were on psychiatric conditions. It should be rephrased.
Reply: We have taken into account the Reviewer’s suggestion and have now slightly modified the title from “Sex Differences in Tryptophan Metabolism: A systematic review focused on neurological disorders” to “Sex Differences in Tryptophan Metabolism: A systematic review focused on neuropsychiatric disorders” (pag. 1, lines 1-3).
2 - In line with the previous comment, mood disorders, anxiety, ADHD, AUD, are psychiatric conditions. This is not clear over the manuscript and it should be clearly differentiated whenever the authors are discussing neurological disorders (as dementia, epilepsy…) or psychiatric disorders (as the remaining discussed on the article).
Reply: We would like to thank the Reviewer this commentary. Considering that, we made some changes to clarify this issue both in abstract (pag. 1, lines 14-15,18-23,27,28) and along manuscript (pag. 3, lines 75,78,82,84,97; pag. 7, line 246,254; pag. 10, line 239,343-350; pag. 16, line 373,374,378-380,384,390,391; pag. 17, line 408,409,417,418; pag. 18, line 458-463; pag. 23, lines 582-594). Additionally, we also altered Figure 3 and its caption to cover neuropsychiatric and not just neurological diseases (pag. 24, line 601,602,606,610).
3 - Chronic fatigue syndrome is not a neurological condition and I’m not sure it is considered a psychiatric disorder.
Reply: We would like to acknowledge the Reviewer´s comment. In fact, there is no consensus agreement amongst medical professionals on how to classify chronic fatigue syndrome (CFS). The World Health Organization (WHO) classifies CFS as a “diseases of the nervous system” but have been some proposals to change that classification to “certain infectious or parasitic diseases”. In response to the distinct proposals, WHO conducted an extensive literature review of research relating to chronic fatigue. As a result of that research, there is no change to the current placement of the term and so, currently, CFS is classified by WHO as a neurological illness and that is why we consider it as a neurological disease in this manuscript.
4 - Methods: - Could the authors provide the reasons for excluding the 1423 articles, by categories?
Reply: We would like to thank the Reviewer for observing missing information regarding the exclusion criteria used. We have added this information to section Methods and created a subsection “Study Eligibility” (pag. 9, lines 314-322 and pag. 10, lines 323-327). So, we decided that exclusion criteria were: 1) Studies on non-human subjects; 2) Studies only including one of the two biological sexes (female or male); 3) Case reports, literature reviews, systematic review and meta-analyses; 4) Studies not published in English; 5) Studies older than 20 years.
5 - Methods - What do the authors mean by “no relevant data” for excluding 23 articles? What were the criteria used?
Response 5: We would like to acknowledge the Reviewer’s comment. The articles referred to as “no relevant data” did not allow to draw conclusions on sexual dimorphisms, even though they include both sexes in their study. That happened because the two sexes were not divided as independent groups and/or direct statistical comparison between sexes was not provided. To further clarification, we have edited the caption of Figure 2 to include this information (pag. 11, lines 357-360).
Reviewer 2 Report
The present manuscript entitled "Sex Differences in Tryptophan Metabolism: A systematic review focused on neurological disorders" is a pertinent review of the literature available to date, since few works have addressed this issue. However, a few details should be addressed.
One of the objectives established for this review included the discussion of the current understanding of how sex differences on Tryp influence the onset and progression of neurological diseases, however, only a few details and examples of neurological diseases are presented. It would be desirable to include a table that better describes the clinical evidence on this point.
Some sentences or ideas in the text appear to be truncated or the explanation is not entirely clear, i.e. line 179
Formatting errors including misplaced commas i.e. line 183 or excessive spacing (line 64-68) should be also improved.
Author Response
Reply to Reviewer #2
1 - One of the objectives established for this review included the discussion of the current understanding of how sex differences on Tryp influence the onset and progression of neurological diseases, however, only a few details and examples of neurological diseases are presented. It would be desirable to include a table that better describes the clinical evidence on this point.
Reply: We have considered the Reviewer’s suggestion. Unfortunately, there is still no direct link between sex differences in tryptophan metabolism and the onset and progression of neuropsychiatric diseases. Although, some sexual dimorphisms in this amino acid metabolism for some neuropsychiatric conditions are consistent with the dysfunction of tryptophan metabolism and its relation to disease severity. So, we have added a new subsection, “Current understanding of how sexual dimorphisms on tryptophan metabolism may be involved in the onset and progression of neuropsychiatric diseases”, to the section Results that includes a summary Table (Table 7) of the current data available on that topic (pag. 19-22).
2 - Some sentences or ideas in the text appear to be truncated or the explanation is not entirely clear, i.e. line 179
Reply: We have now checked the manuscript and improved them to clarify some less explicit phrases.
3 - Formatting errors including misplaced commas i.e. line 183 or excessive spacing (line 64-68) should be also improved.
Reply: We considered this comment of Reviewer and we have changed that sentence to improve the misplaced comma (pag. 6, lines 193). The manuscript has been reviewed in an attempt to find more similar errors, but that has not been confirmed. Regarding the excessive spacing shown in the caption of Figure 1, it is not excessive spacing between words but rather a result of text justification. As the formatting tools do not allow the split of the links in that sentence, this leads to the appearance of a larger spacing.